# The miR-100-5p Targets SMARCA5 to Regulate the Apoptosis and Intracellular Survival of BCG in Infected THP-1 Cells

**DOI:** 10.3390/cells12030476

**Published:** 2023-02-01

**Authors:** Li Su, Tingting Zhu, Han Liu, Yifan Zhu, Yongchong Peng, Tian Tang, Shiying Zhou, Changmin Hu, Huanchun Chen, Aizhen Guo, Yingyu Chen

**Affiliations:** 1The State Key Laboratory of Agricultural Microbiology, College of Veterinary Medicine, Huazhong Agricultural University, Wuhan 430070, China; 2Key Laboratory of Development of Veterinary Diagnostic Products, Ministry of Agriculture and Rural Affairs, Huazhong Agriculture University, Wuhan 430070, China; 3National Professional Laboratory for Animal Tuberculosis (Wuhan), Ministry of Agriculture and Rural Affairs, Huazhong Agriculture University, Wuhan 430070, China; 4Hubei International Scientific and Technological Cooperation Base of Veterinary Epidemiology, Hubei Hongshan Laboratory, Huazhong Agricultural University, Wuhan 430070, China

**Keywords:** miR-100-5p, BCG, SMARCA5, THP-1, apoptosis, survival

## Abstract

*Mycobacterium tuberculosis* (*M. tb*) is the causative agent of tuberculosis (TB) that leads to millions of deaths each year. Extensive evidence has explored the involvement of microRNAs (miRNAs) in *M. tb* infection. Limitedly, the concrete function of microRNA-100-5p (miR-100-5p) in *M. tb* remains unexplored and largely elusive. In this study, using Bacillus Calmette–Guérin (BCG) as the model strain, we validated that miR-100-5p was significantly decreased in BCG-infected THP-1 cells. miR-100-5p inhibition effectively facilitated the apoptosis of infected THP-1 cells and reduced BCG survival by regulating the phosphatidylinositol 3-kinase/AKT pathway. Further, SMARCA5 was the target of miR-100-5p and reduced after miR-100-5p overexpression. Since BCG infection down-regulated miR-100-5p in THP-1 cells, the SMARCA5 expression was up-regulated, which in turn increased apoptosis through caspase-3 and Bcl-2 and, thereby, reducing BCG intracellular survival. Collectively, the study uncovered a new molecular mechanism of macrophage to suppress mycobacterial infection through miR-100-5p and SMARCA5 pathway.

## 1. Introduction

*Mycobacterium tuberculosis* (*M. tb*) is the leading etiological agent of tuberculosis (TB) in humans, with 9.9 million new cases and 1.5 million mortalities in 2020 worldwide [1]. Although one-third of the population has been infected [2], only about 10% possibly become active infection. In the other 90% of the latently infected population, this pathogen remains dormant by escaping clearance of host cells but could recover to be active form and cause TB at an unpredictable time. Therefore, it is ideal to clear the intracellular *M. tb*. Macrophages are the main host cells of *M. tb*. Although it is known that the macrophages can develop various strategies to kill *M. tb*, such as modulating cell death programs, regulating inflammatory responses, and inducing autophagy, during *M. tb* infection [3,4], *M. tb* can persist in the cells by overcoming the microbicidal mechanisms of macrophages. Therefore, more studies are needed to dissolve the mystery of *M. tb* intracellular survival. 

Apoptosis is one of the important strategies of host resistance to *M. tb* and response to bacilli survival and transmission [5,6]. Studies have shown that host-mediated apoptosis leads to apoptosis and reduce the number of bacteria and the effective cross-presentation of the bacterial antigen [7,8]. When infected with mycobacteria, attenuated virulent strains, such as Bacillus Calmette–Guérin (BCG) and *M. tb H37Ra*, can better induce apoptosis than strong strains [9].

MicroRNA (miRNA) is an important posttranscriptional regulatory factor that regulate multiple target genes with related functions [10,11]. miR-155 targets FOXO3, inhibits apoptosis, and helps bacteria escape the immune response when infected with BCG [12]. miR-146a can inhibit the *M. tb*-mediated inflammatory response through the IRAK-1/TRAF-6 pathway and promote bacterial proliferation in RAW264.7 macrophages [13]. miR-21-5p increased in the *M. tb*-infected THP-1 or RAW264.7 cells and induces apoptosis by targeting Bcl-2 and Toll-like receptor (TLR) 4 [14]. 

MiR-100-5p is a highly conserved miRNA that can promote autophagy activation by targeting mTOR, decrease apoptosis by binding to TLR8, affect the proliferation, migration, and invasion by targeting IGF1R in many malignant tumors [15,16,17,18] and cardiac hypertrophy [19]. However, its role in *M. tb* infection is still unknown. 

Therefore, the study aim to investigate the functions of miR-100-5p during *M. tb* infection using the BCG as the reference strain. We demonstrated that miR-100-5p was downregulated after BCG infection, targeted to SMARCA5, inhibited the phosphatidylinositol 3-kinase (PI3K)/AKT signaling pathway to facilitate cell apoptosis, and inhibited BCG in macrophages. Our findings expand the current understanding of the interaction mechanism between *M. tb* and macrophage, and miR-100-5p could be considered as potential targets for TB treatment.

## 2. Materials and Methods

### 2.1. Bacterial Growth and Cell Culture

*Mycobacterium bovis* BCG (Tokyo strain; ATCC 35737) was kindly presented by Professor Liu Junyan (Wuhan University, Wuhan, China). BCG was grown in Middlebrook 7H9 broth medium (BD, Franklin, NJ, USA) with 10% oleic acid, albumin, dextrose, and catalase (OADC; BD), 0.5% glycerol (Sigma-Aldrich, St. Louis, MO, USA), and 0.05% Tween-80 (Sigma-Aldrich) at 37 °C, or 7H11 agar plates (BD) with 10% OADC and 0.5% glycerol at 37 °C.

Human acute monocytic leukemia cell line THP-1 (ATCC TIB-202) was cultured in RPMI 1640 medium (HyClone, Logan, UT, USA) with 10% fetal bovine serum (FBS; Gibco, Carlsbad, CA, USA) at 37 °C in 5% CO_2_. Before infection, THP-1 cells were seeded on a 12-well plate and stimulated with 40 ng/mL phorbol myristate acetate (PMA; Sigma-Aldrich) for 12 h. Human embryonic kidney cell line HEK 293T (ATCC CRL-3216) was grown in Dulbecco’s modified Eagle’s medium (HyClone) with 10% FBS at 37 °C in 5% CO_2_.

### 2.2. Cell Infection with BCG

Bacterial cultures at the logarithmic growth stage were centrifuged at 3800× *g* for 10 min. The precipitates were collected and resuspended in Hank’s balanced salt solution (Gibco) and dispersed into a single-cell suspension. The concentrations were estimated by OD_600_ values [20,21]. THP-1 cells were infected with BCG at a multiplicity of infection (MOI) of 10 for 8 h [21,22], followed by washing with 1 mL of 10% FBS medium plus 100 μg/mL gentamycin. This time point was set as 0 h. The samples were collected at the designated time points.

### 2.3. Total RNA Extraction and Sequencing

Total RNA was extracted and sequenced as previously described [23]. Briefly, total RNA was isolated using TRIzol reagent (Invitrogen, Carlsbad, CA, USA) in accordance with the instructions of the manufacturer. RNA purity and integrity were checked by agarose gel electrophoresis and quantified by determining the absorbance of A260 on SmartSpec (Bio-Rad, Hercules, CA, USA). A Balancer NGS Library Preparation Kit for small RNA/miRNA (GnomeGen, San Diego, CA, USA) was used for small RNA cDNA library preparation, in accordance with the instructions of the manufacturer. The purified small RNA libraries were quantified for cluster generation and 36 nt single-end sequencing analysis using Illumina GAIIx (Illumina, San Diego, CA, USA).

### 2.4. Cell Transfection

THP-1 cells were plated on 12-well plates (1 × 10^6^ cells/well) for 12 h before transfection, as reported previously [21]. Synthesized RNA fragments, including miR-100-5p mimic, miR-100-5p inhibitor, siSMARCA5, and controls, were mixed with jetPRIME^®^ transfection reagent (Polyplus, Illkirch, France) for 10 min. The mixture (100 μL/well) was added to cells and cultured at 37 °C in 5% CO_2_ for 24 h. miR-100-5p mimic, miR-100-5p inhibitor, siSMARCA5, and the respective controls were synthesized by a commercial company (GenePharma, Suzhou, China). All sequences are listed in Table 1.

### 2.5. Quantitative Real-Time Polymerase Chain Reaction

The miR-100-5p and SMARCA5 expression was determined by quantitative real-time polymerase chain reaction (qRT-PCR). Total RNA was extracted using TRIzol reagent (Invitrogen, Carlsbad, CA, USA) following the manufacturer’s instructions. The concentration was detected by NanoDrop 2000 (Thermo Fisher, Waltham, MA, USA). Total RNA (1 µg) was used to synthesize cDNA using the All-in-One^TM^ miRNA qRT-PCR Detection Kit (for miRNA; Vazyme, Nanjing, China) or HiScript^®^ II qRT SuperMix (for mRNA; Vazyme). After reverse transcription, qRT-PCR was used to detect miRNA and mRNA expression by AceQ qPCR SYBR Green Master Mix on an ABI ViiA 7 Real-time PCR System (Applied Biosystems, Carlsbad, CA, USA). miRNA was normalized to U6 and mRNA was normalized to β-actin. The 2^−ΔΔCt^ method was used to calculate the relative expression. All primers are listed in Appendix A.

### 2.6. Cell Death Assay

Cell death was tested by assaying the medium lactate dehydrogenase (LDH) content by the LDH Release Assay Kit (Beyotime, Shanghai, China). At the designated time points, the medium was incubated with a detection reagent in the dark for 30 min and LDH values were recorded at OD_450_.

### 2.7. Apoptosis Assay

Apoptosis was evaluated using the Annexin V-Fluorescein Isothiocyanate/Propidium Iodide (PI) Apoptosis Detection Kit (Vazyme) following the manufacturer’s instructions. Briefly, the supernatant was discarded, and cells were washed once with precooled phosphate-buffered saline (PBS; HyClone). Cells were digested with trypsin without EDTA. The precipitates were collected by centrifugation at 800× *g* for 5 min at 4 °C. Cells were washed using PBS, centrifuged again, and resuspended using 1× binding buffer. Finally, cells were stained using PI and Annexin V in the dark. Flow cytometry (CytoFLEX; Beckman Coulter, Brea, CA, USA) was used to analyze the stained cells.

### 2.8. Western Blotting Assay

Cells were washed twice with ice-cold PBS and scrapped in radioimmunoprecipitation assay buffer with protease and phosphatase inhibitors. Total protein samples were electrophoresed on 10% sodium dodecyl sulfate-polyacrylamide gel electrophoresis and transferred to polyvinylidene fluoride membranes. All membranes were incubated overnight with the appropriate antibodies against anti-caspase-3 antibody (1:5000; ab32351; Abcam, Boston, MA, USA), anti-Bcl-2 antibody (1:1000; SZ10-03; Huabio, Hangzhou, China), mouse monoclonal antibody to β-actin (1:2000; 60008-1-Ig; Proteintech, Chicago, IL, USA), mouse monoclonal antibody to glyceraldehyde 3-phosphate dehydrogenase (GAPDH; 1:20,000; 60004-1-Ig; Proteintech), anti-SNF2H (SMARCA5) antibody (1:1000; ab183730; Abcam), anti-PI3K p85 α-antibody (1:1000; ab191606; Abcam), anti-PI3K catalytic subunit γ/PI3K γ-antibody (1:1000; ab32089; Abcam), anti-pan-AKT antibody (1:500; ab8805; Abcam), and anti-pan-AKT (phospho-T308) antibody (1:1000; ab38449; Abcam). Proteins were visualized using the appropriate secondary antibodies conjugated horseradish peroxidase-based chemiluminescence detection system. Protein band intensities were determined using the Western Bright ECL kit (Advansta, Menlo Park, CA, USA) and analyzed using ImageJ.

### 2.9. Assay on Intracellular Bacterial Number

THP-1 cells were infected with BCG at an MOI of 10, as described previously, to assess bacterial survival in vitro [21]. THP-1 cells were lysed with 1 mL sterile distilled water at the indicated time points. Tenfold serial dilutions of the lysates were plated on 7H11 agar plates and the colony-forming unit (CFU) was counted 21 days later.

### 2.10. Target Gene Prediction and Analysis

TargetScan 7.1 (http://www.targetscan.org, accessed on 20 June 2020), miRBD (http://mirdb.org/miRDB/, accessed on 20 June 2020), miR-Walk (http://mirwalk.umm.uni-heidelberg.de/, accessed on 20 June 2020), StarBase (https://starbase.sysu.edu.cn/index.php, accessed on 20 June 2020), and miRpathDB (https://mpd.bioinf.uni-sb.de/, accessed on 20 June 2020) were used to analyze the putative target genes of miR-100-5p. The predicted results were analyzed by Venn diagram to find the intersection (http://bioinformatics.psb.ugent.be/webtools/Venn/, accessed on 20 June 2020). Putative targets predicted by the five software were used for further screening (Appendix A).

### 2.11. Dual-Luciferase Reporter Assay

Wild-type (WT) and mutant (mut) 3′-untranslated region (UTR) of SMARCA5, MTMR3, HS3ST3B1, TRIB2, and EPDR1, which contains the predicted miR-100-5p binding site, were cloned into the psiCHECK-2 luciferase reporter vector (Promega, Madison, WI, USA) to obtain the SMARCA5/MTMR3/HS3ST3B1/TRIB2/EPDR1 3′-UTR-luc construct. The constructs were cotransfected with miR-100-5p mimic or NC mimic into HEK 293T cells. All primers are listed in Appendix A. A luciferase assay was performed using the Dual-Luciferase Reporter Assay System (Promega) 24 h after cotransfection.

### 2.12. Statistical Analysis

Data were the mean ± standard error of the mean (SEM). Statistical significance was calculated using the Student’s *t*-test for one comparison and analysis of variance (ANOVA) for more than one comparison in GraphPad Prism version 7.0. The *p* < 0.05 was determined to be statistically significant difference, and *p* < 0.05, 0.01, and 0.001 were expressed as *, **, and ***, respectively, in figures.

## 3. Results

### 3.1. BCG Infection Reduced miR-100-5p Expression in THP-1 Cells

RNA-seq data showed that miR-100-5p was downregulated in BCG and *M. tb*-infected THP-1 cells at 6 and 24 h post-infection (Table 2). qRT-PCR also confirmed that in BCG infected THP-1 cells, miR-100-5p was significantly downregulated at MOI 10 at 0, 12, and 24 hpi (hours post infection) (*p* < 0.001) compared with uninfected group (Figure 1). This indicates that BCG can reduce miR-100-5p expression in THP-1 cells.

### 3.2. miR-100-5p Inhibited Apoptosis of BCG Infected THP-1 Cells

To examine the potential function of miR-100-5p during BCG infection, its effect on cytotoxicity was analyzed by the LDH release assay. THP-1 cells were transiently transfected with miR-100-5p mimic or inhibitor (Appendix A). Treatment with miR-100-5p mimic significantly decreased cell death at 0 (*p* < 0.01), 12 (*p* < 0.001), and 48 (*p* < 0.001) hpi (Figure 2A), whereas miR-100-5p inhibitor increased cell death at 12, 24, and 48 hpi (*p* < 0.001; Figure 2B), indicating that miR-100-5p affected THP-1 cell cytotoxicity induced by BCG infection.

Flow cytometry was conducted to better characterize the miR-100-5p cytotoxicity function. In the miR-100-5p mimic-transfected group, THP-1 had significantly lower apoptosis than control (*p* < 0.01), whereas apoptosis increased when THP-1 cells were transfected with miR-100-5p inhibitor (*p* < 0.01; Figure 3A–C). miR-100-5p overexpression markedly reduced the amount of caspase-3 (*p* < 0.001) but enhanced Bcl-2 expression (*p* < 0.001), whereas miR-100-5p inhibitor enhanced caspase-3 expression (*p* < 0.001) but decreased Bcl-2 (*p* < 0.001; Figure 3D–F), indicating that miR-100-5p increases the apoptosis responding to BCG infection.

### 3.3. miR-100-5p Regulates the PI3K/AKT Signalling Pathway

PI3K, p-PI3K, AKT, and p-AKT expression levels were determined to identify the pathway that miR-100-5p regulated apoptosis. Compared to the control mimic treatment group, miR-100-5p mimic promoted p-AKT (*p* < 0.001) and p-PI3K (*p* < 0.05) protein expression in THP-1 cells. Furthermore, anti-miR-100-5p-treated THP-1 cells repressed p-PI3K and p-AKT proteins compared to control inhibitor-treated macrophages (*p* < 0.001; Figure 4A–D), indicating that miR-100-5p can inhibit apoptosis by activating the PI3K/AKT signaling pathway.

### 3.4. miR-100-5p Promoted BCG Intracellular Survival via Suppression of Apoptosis

BCG number was assessed in vitro to investigate whether miR-100-5p-dependent repression apoptosis enhances BCG survival. THP-1 cells were infected with BCG for 48 h at 24 h after transfection of miR-100-5p mimic or control mimic. Compared with the control mimic group, the miR-100-5p mimic group had significantly higher CFU (*p* < 0.001) counts (Figure 5A) and the CFU counts in the miR-100-5p inhibitor group were significantly less than in the control group (*p* < 0.01; Figure 5B). A ratio analysis based on 0 h also proved the above results (Figure 5C), indicating that miR-100-5p can promote BCG intracellular survival in macrophages.

### 3.5. miR-100-5p Directly Targets SMARCA5

A series of transcripts were found as potential targets of miR-100-5p using the base alignment approach. Accordingly, seven and 25 genes were predicted by five and four software, respectively (Figure 6A). Among them, SMARCA5, MTMR3, HS3ST3B1, EPDR1, and TRIB2 were chosen for the next step; all of them were found to have specific binding sites for miR-100-5p in its 3′-UTR (Figure 6B and Appendix A). Luciferase reporter constructs were generated by cloning either WT or mut 3′-UTR of targets into a pSicheck vector to confirm whether they are direct targets of miR-100-5p. These plasmids were co-transfected with miR-100-5p mimic into HEK 293T cells and the lysates were analyzed 24 h later. Transfection with miR-100-5p mimic markedly inhibited the luciferase activity for the WT 3′-UTR of all five genes (*p* < 0.01) but showed no significant repressive effects on the mutated 3′-UTR of genes compared to control double-stranded RNA (Figure 6C and Appendix A). These results suggested that miR-100-5p may suppress SMARCA5, MTMR3, HS3ST3B1, EPDR1, and TRIB2 expression by binding to the 3′-UTR in a direct and sequence-specific manner.

SMARCA5, MTMR3, HS3ST3B1, EPDR1, and TRIB2 mRNA levels were much higher in BCG-infected cells than in their representative controls (Figure 6D and Appendix A). THP-1 cells were transfected with miR-100-5p mimic or inhibitor and infected with BCG to examine the effects of miR-100-5p on SMARCA5, MTMR3, HS3ST3B1, EPDR1, and TRIB2 expression during BCG infection. However, only SMARCA5 showed stable results. Treatment with miR-100-5p mimic significantly inhibited SMARCA5 at both mRNA and protein levels, whereas miR-100-5p inhibitor markedly enhanced SMARCA5 at mRNA and protein levels in BCG-infected macrophages (Figure 6E–H). These results suggested that miR-100-5p expression suppresses SMARCA5 expression during BCG infection.

### 3.6. miR-100-5p Regulates Apoptosis and BCG Survival in THP-1 via SMARCA5

THP-1 cells were transfected with miR-100-5p inhibitor or siRNA SMARCA5 and infected with BCG for 24 h to determine whether miR-100-5p regulates apoptosis and BCG intracellular survival by targeting SMARCA5. Compared to the NC inhibitor + control siRNA control group, SMARCA5 expression was remarkably reduced in the NC inhibitor + siSMARCA5 group (*p* < 0.01) and increased in the miR-100-5p inhibitor + control siRNA group (*p* < 0.001). SMARCA5 levels were significantly lower in the miR-100-5p inhibitor + siSMARCA5 group than in the miR-100-5p inhibitor + control siRNA group (*p* < 0.001; Figure 7).

Based on the SMARCA5 interference model, LDH release assay, flow cytometry, and Western blotting were performed to detect apoptosis and CFU assay was performed to detect intracellular bacterial load.

Compared to the control group (NC inhibitor + control siRNA) with miR-100-5p inhibition only, LDH release increased significantly at 48 hpi (*p* < 0.001; Figure 8A), the apoptosis rate of flow cytometry was promoted (*p* < 0.05; Figure 8B,C), and caspase-3 expression increased, but Bcl-2 was significantly reduced (*p* < 0.001; Figure 8D–F). However, based on miR-100-5p inhibition, when SMARCA5 was silenced (miR-100-5p inhibitor + siSMARCA5), LDH release (Figure 8A), apoptosis rate (Figure 8B,C), and caspase-3 expression decreased, whereas Bcl-2 increased than in the control group (miR-100-5p inhibitor + control SMARCA5; *p* < 0.001; Figure 8D–F). Furthermore, RNA interference of SMARCA5 increased BCG intracellular survival in THP-1 cells (Figure 8G,H). These findings demonstrated that miR-100-5p directly targets SMARCA5, resulting in the potent inhibition of apoptosis and cumulative effects on survival of intracellular BCG.

## 4. Discussion

In this study, we found that miR-100-5p regulates SMARCA5 and, thus, mediates cell apoptosis and BCG survival in cells. *M. tb* is a persistent intracellular pathogen that survives within the hostile microenvironment of the macrophage. During the infection, host cells triggers a series immunity responds to destroy the pathogen and clear the debris including apoptosis [24], which prevents the bacterial release and the spreading of mycobacterial infection and activating the host’s innate and adaptive immune response [25].

miR-100-5p is involved in multiple diseases and affects apoptosis in previous studies. In chordoma cells, miR-100-5p suppresses proliferation and induces apoptosis via insulin-like growth factor-I receptor [26]. Oncogenic miR-100-5p is associated with cellular viability, migration, and apoptosis in renal cell carcinoma and can be used as a diagnostic biomarker for renal cell carcinoma [27]. In goat endometrial interstitial cells, miR-100-5p can inhibit cell proliferation and promote apoptosis through its target HOXA1 and the molecular sponge circ-9110 [28]. Cortical neurons can elicit cell-autonomous apoptosis through miR-100-5p-induced TLR activation and contribute to neurodegeneration [15]. In the current study, we demonstrated a sharp low level of miR-100-5p in BCG-infected THP-1 cells and identified that miR-100-5p also regulate apoptosis in *M. tb* infection. This is in agreement with the function of miR-100-5p in other diseases.

As reported, miR-100-5p has many targets in different diseases, such as mTOR [19], IGF1R [26], BMPR2 [29], and SMARCA5 [30]. To our knowledge, the function of miR-100-5p and SMARCA5 in mycobacterium have not previously been reported. Here, we found that SMARCA5 was regulated by miR-100-5p using bioinformatics prediction and a dual-luciferase reporter system during *M. tb* infection. SMARCA5 is an ATPase from the ISWI subfamily and acts as a molecular motor for assembling and sliding nuclear complexes of nucleosomes of basic chromatin subunits. The complex containing SMARCA5 can guide the transcription of ribosomes in NORC and B-WICH complexes and participate in the DNA repair mechanism to coordinate the formation of the high-order chromatin structure of centromeres and chromosomes [31]. Previous studies reported that SMARCA5 can affect the cell apoptosis process [32,33], lack of SMARCA5 reduced the development of thymocyte DN3 stage and proB stage of early B cells, and increased cleaved caspase-3 to promote apoptosis [31]. In this study, through cotransfection miR-100-5p and siSMARCA5, miR-100-5p negatively regulated SMARCA5 and affected apoptosis by targeting SMARCA5. This was in agreement with the previous studies. Meanwhile, SMARCA5 could also regulate DNA damage [34], which can lead to growth arrest and apoptosis [35,36]. Therefore, it was speculated that apoptosis regulated by miR-100-5p through SMARCA5 may be related to DNA damage caused by pathogen infections.

Besides apoptosis, miRNAs can also affect bacterial intracellular survival. In *M. tb* infection, miR-27a induction can inhibit autophagy from promoting the intracellular survival in macrophages [37]. This study also found that miR-100-5p can lead to the accumulation of intracellular bacterium. When infected with BCG, the bacterium tried to establish a persistent infection. As the host cells began their defense system to break the persistent infection, it induced apoptosis and decreased the intracellular ability of BCG by inhibiting miR-100-5p.

PI3K/AKT signaling pathway plays a major role in cell metabolism, proliferation, and apoptosis [38]. The activated PI3K/AKT pathway can increase the activity of cytochrome c oxidase to inhibit apoptosis [39]. miRNA-regulated apoptosis through the PI3K/AKT pathway has attracted extensive attention previously, such that miR-107 aggravates apoptosis by inactivating the PI3K/AKT signaling pathway by targeting FGF9 [40]. miR-100 overexpressed in osteosarcoma could decrease IGF receptor expression and inhibit PI3K/AKT signal transduction and mitogen-activated protein kinase/extracellular signal-regulated kinase pathways from promoting apoptosis [41]. This study identified that PI3K and AKT could be phosphorylated to activate the PI3K/AKT pathway by overexpressing miR-100-5p. However, the detailed mechanism of the miR-100-5p-suppressed PI3K/AKT pathway in regulating apoptosis remains unclear. Insufficient understanding of how miR-100-5p is downregulated in BCG-infected macrophages was also noted. These are worthy of further exploration.

## 5. Conclusions

These findings indicated that, after BCG infection of THP-1 cells, miR-100-5p was downregulated and targeted to SMARCA5 and inactivated the PI3K/AKT pathway, resulting in apoptosis and ultimately reducing BCG intracellular survival in macrophages (Figure 9). This finding offers a new insight of further understanding of host defense mechanism and supports the development of host-directed anti-TB therapeutic approaches.

## Figures and Tables

**Figure 1 cells-12-00476-f001:**
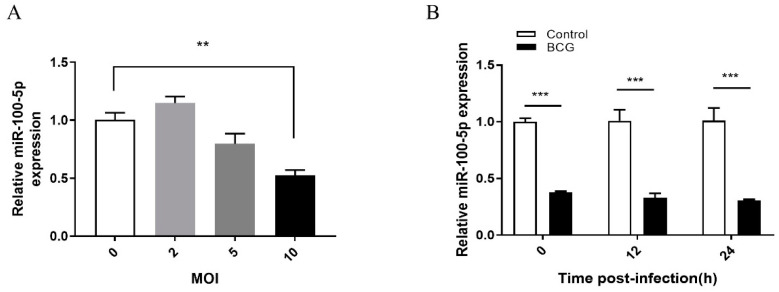
miR-100-5p expression in BCG-infected THP-1 cells. (**A**) PMA-differentiated THP-1 cells were infected with BCG at MOIs of 2, 5, and 10 for 8 h. Total RNA was extracted at 24 hpi. The miR-100-5p expression level was measured by qRT-PCR normalized with U6 as the internal reference. (**B**) PMA-differentiated THP-1 cells were infected with BCG at an MOI of 10 for 8 h. Total RNA was extracted at 0, 12, and 24 hpi. The miR-100-5p expression level was measured by qRT-PCR normalized with U6 as the internal reference. Data are representative of three independent experiments with biological duplicates in each ((**A**,**B**); mean ± s.e.m. for *n* = 3 duplicates). ** *p* < 0.01; *** *p* < 0.001 (ANOVA).

**Figure 2 cells-12-00476-f002:**
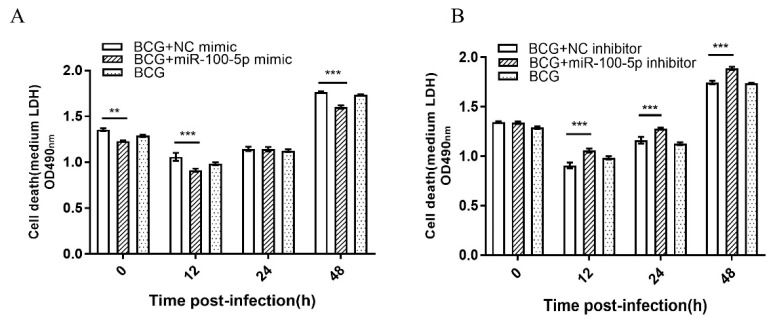
miR-100-5p inhibits LDH release after BCG infection. LDH assay in THP-1 cells transfected with NC mimic, miR-100-5p mimic (**A**) or NC inhibitor, miR-100-5p inhibitor (**B**) and infected with BCG (MOI = 10) for 8 h. Cell death was detected by medium LDH release assay at 0, 12, 24, and 48 hpi. Data are from three independent experiments with biological duplicates in each ((**A**,**B**); mean ± s.e.m. for *n* = 3 duplicates). ** *p* < 0.01; *** *p* < 0.001.

**Figure 3 cells-12-00476-f003:**
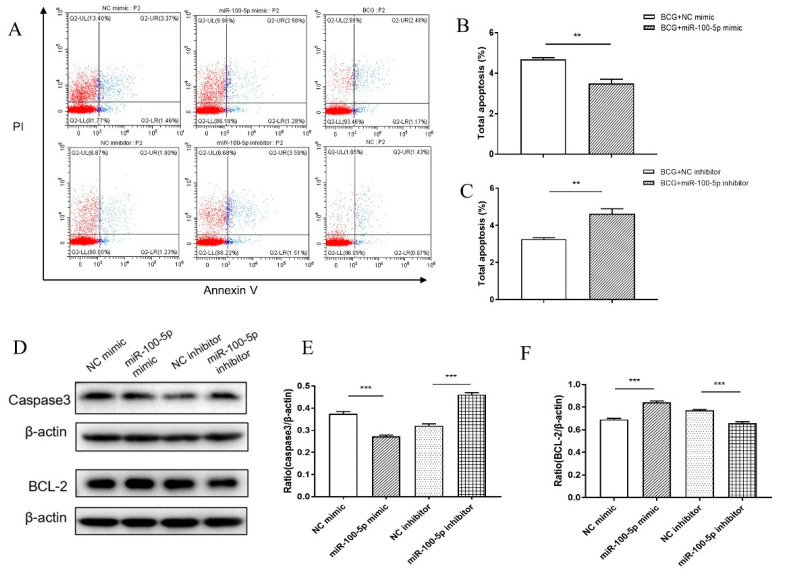
Regulation of apoptosis of BCG-infected THP-1 cells by miR-100-5p. THP-1 cells were transfected with NC mimic, mimic, NC inhibitor, or miR-100-5p inhibitor and then infected with BCG (MOI = 10) for 8 h. (**A**) Cell apoptosis was detected by flow cytometry at 12 hpi. There were six groups: transfected NC mimic and infected BCG, transfected miR-100-5p mimic and infected BCG, transfected NC inhibitor and infected BCG, transfected miR-100-5p inhibitor and infected BCG, untransfected but infected BCG (BCG), and untransfected and uninfected (NC). (**B**,**C**) Cell total apoptosis rate with different groups. (**D**) Total proteins were collected at 12 hpi. Caspase-3 and Bcl-2 protein levels were detected by Western blotting assay. (**E**,**F**) Relative protein expression ratios were analyzed using ImageJ with normalization to β-actin. Data are from three independent experiments with biological duplicates in each ((**A**–**C**); mean ± s.e.m. of *n* = 3 duplicates) or representative of three independent experiments (**D**–**F**). ** *p <* 0.01; *** *p* < 0.001 (Student’s *t*-test for one comparison or ANOVA for more than one comparison).

**Figure 4 cells-12-00476-f004:**
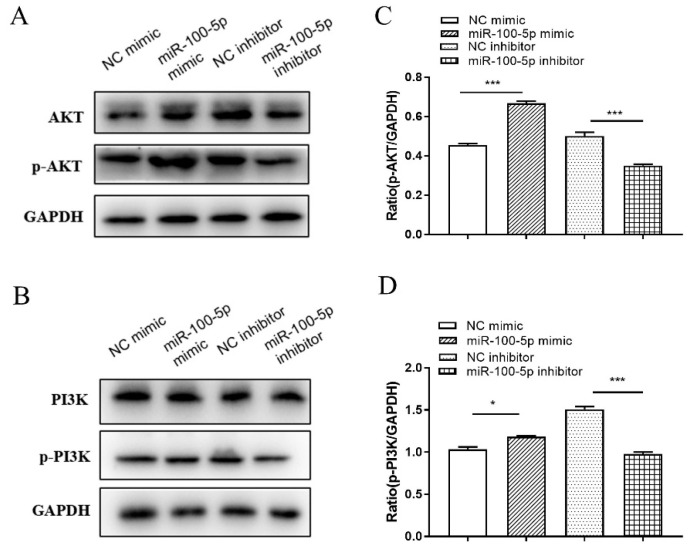
miR-100-5p increases PI3K and AKT phosphorylation. THP-1 cells were transfected with NC mimic, miR-100-5p mimic, NC inhibitor, or miR-100-5p inhibitor for 24 h and infected with BCG at an MOI of 10 for 8 h. (**A**,**B**) Total proteins were collected at 12 hpi. p-PI3K and p-AKT protein levels were detected by Western blotting assay. (**C**,**D**) Relative protein expression ratios were analyzed using ImageJ with normalization to GAPDH. Data are representative of three independent experiments ((**C**,**D**), mean ± s.e.m.). * *p <* 0.05; *** *p <* 0.001.

**Figure 5 cells-12-00476-f005:**
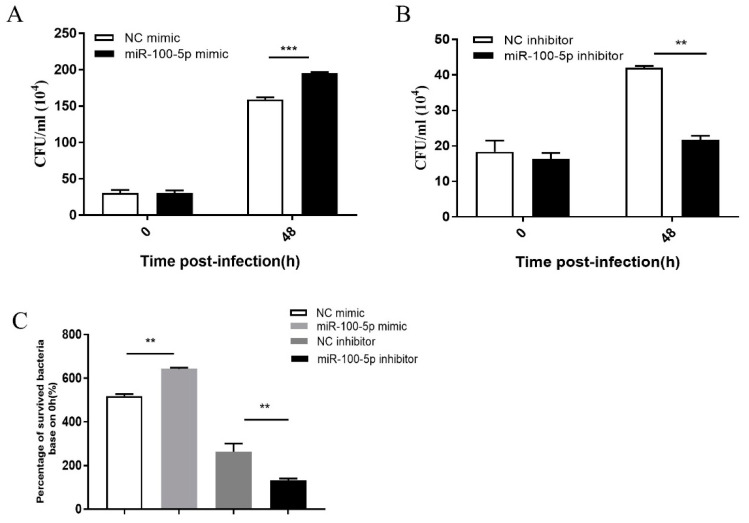
miR-100-5p promotes BCG survival in macrophages. THP-1 cells were transfected with NC mimic, miR-100-5p mimic, NC inhibitor, or miR-100-5p inhibitor for 24 h and infected with BCG at MOI 10 for 8 h. (**A**,**B**) Viable intracellular BCG in THP-1 cells transfected with miR-100-5p mimic or inhibitor compared to the negative control. (**C**) Percentage analysis of surviving intracellular bacteria based on 0 hpi with four groups. Data are from three independent experiments with biological duplicates in each ((**A**–**C**); mean ± s.e.m. of *n* = 3 duplicates). ** *p <* 0.01; *** *p <* 0.001.

**Figure 6 cells-12-00476-f006:**
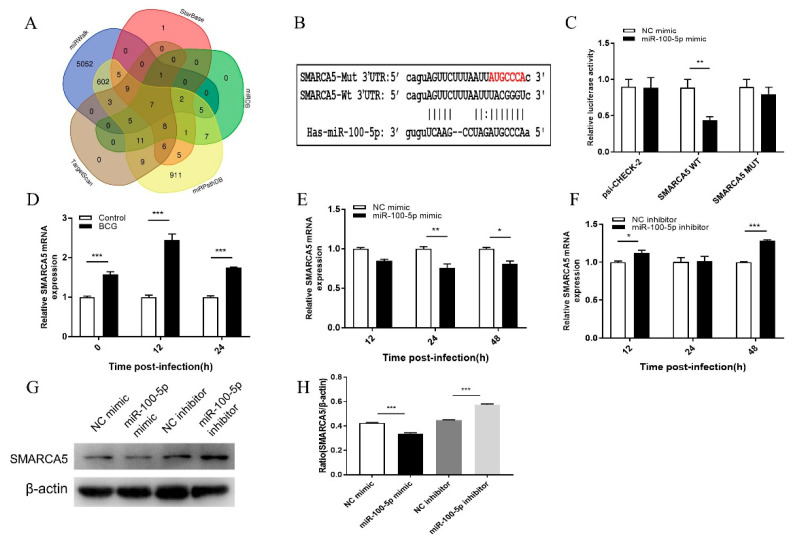
Screening and validation of miR-100-5p target genes. (**A**) miR-100-5p target gene prediction with five software and analyzed by Venn diagram. (**B**) Binding site of miR-100-5p with SMARCA5 WT or mut 3′-UTR as predicted by online prediction software. (**C**) Dual-luciferase reporter assay performed in HEK 293T cells. miR-100-5p mimic or NC mimic was cotransfected with dual-luciferase reporter plasmids, such as psiCHECK2 control, psiCHECK2-SMARCA5-wt, and psiCHECK-SMARCA5-mut. The ratio represents the Renilla to firefly (Rluc/Fluc) ratio in miR-100-5p mimic versus NC mimic. (**D**) Differential SMARCA5 expression between BCG-infected (MOI = 10) and uninfected cells measured by qRT-PCR at 0, 12, and 24 hpi. (**E**,**F**) THP-1 cells were transfected with NC mimic, miR-100-5p mimic, NC inhibitor, or miR-100-5p inhibitor for 24 h and infected with BCG at an MOI 10 for 8 h. SMARCA5 mRNA expression was detected by qRT-PCR at 12, 24, and 48 hpi. (**G**–**H**) SMARCA5 protein expression detected by Western blotting assay. The expression ratio was analyzed using ImageJ with normalization to β-actin. Data are from three independent experiments with biological duplicates in each ((**C**–**F**); mean ± s.e.m. of *n* = 3 duplicates) or representative of three independent experiments ((**G**,**H**), mean ± s.e.m). * *p* < 0.05; ** *p* < 0.01; ****p* < 0.001.

**Figure 7 cells-12-00476-f007:**
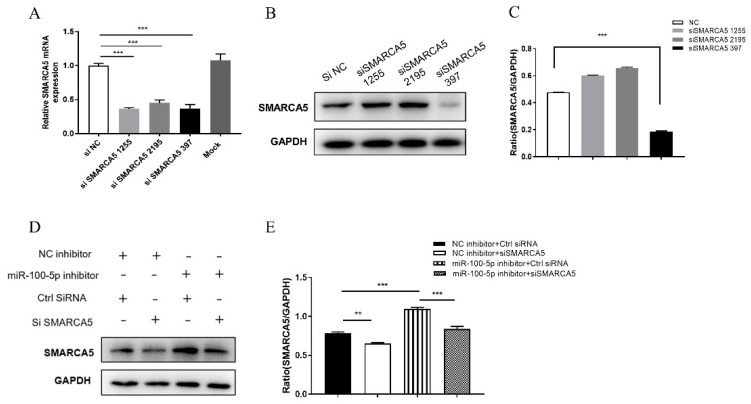
Construction of miR-100-5p and siSMARCA5 cotransfection model. (**A**–**C**) After transfection of three siRNAs, SMARCA5 mRNA and protein expression levels were detected by qRT-PCR and Western blotting. The protein expression ratios were analyzed using ImageJ with normalization to GAPDH. (**D**) Western blot detected SMARCA5 protein expression from THP-1 cells pretreated with miR-100-5p inhibitor, siSMARC5 and their negative controls infected with BCG. (**E**) Expression ratios analyzed using ImageJ with normalization to GAPDH. Data are from three independent experiments with biological duplicates in each ((**A**); mean ± s.e.m. of *n* = 3 duplicates) or representative of three independent experiments ((**B**–**E**), mean ± s.e.m). ** *p* < 0.01; *** *p* < 0.001.

**Figure 8 cells-12-00476-f008:**
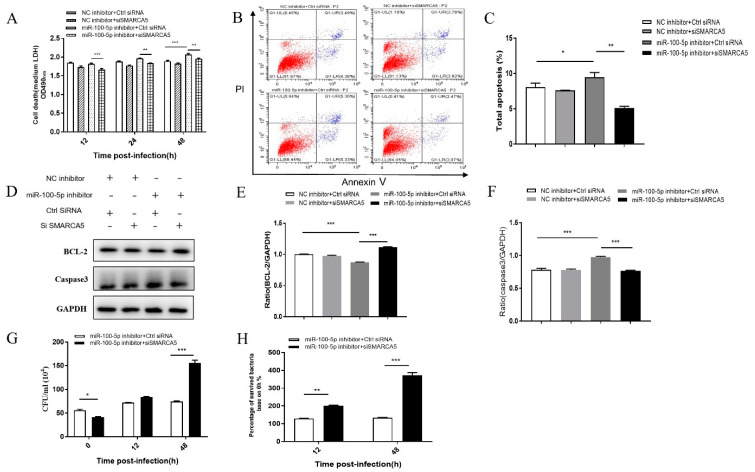
miR-100-5p inhibits apoptosis and promotes BCG survival via SMARCA5 in BCG-infected macrophages. THP-1 cells were cotransfected with NC inhibitor, miR-100-5p inhibitor, siSMARCA5, or control siRNA for 24 h and infected with BCG at an MOI 10 for 8 h. There were four groups: cotransfected NC inhibitor and siRNA control, cotransfected NC inhibitor and siSMARCA5, cotransfected miR-100-5p inhibitor and siRNA control, and cotransfected miR-100-5p inhibitor and siSMARCA5. (**A**) Medium LDH release detected at 12, 24, and 48 hpi with OD490 nm. (**B**,**C**) Cell apoptosis identified by flow cytometry at 12 hpi. The total apoptosis rate was analyzed by GraphPad Prism. (**D**) Caspase-3 and Bcl-2 protein levels detected by Western blotting assay. (**E**,**F**) Relative protein expression ratios analyzed using ImageJ with normalization to GAPDH. (**G**,**H**) Number of intracellular bacteria (CFU/mL) determined at 48 hpi. The percentage of surviving intracellular bacteria based on 0 hpi was analyzed. Data are from three independent experiments with biological duplicates in each ((**A**–**C**,**G**,**H**); mean ± s.e.m. of *n* = 3 duplicates) or representative of three independent experiments ((**D**–**F**), mean ± s.e.m). * *p <* 0.05; ** *p <* 0.01; *** *p* < 0.001.

**Figure 9 cells-12-00476-f009:**
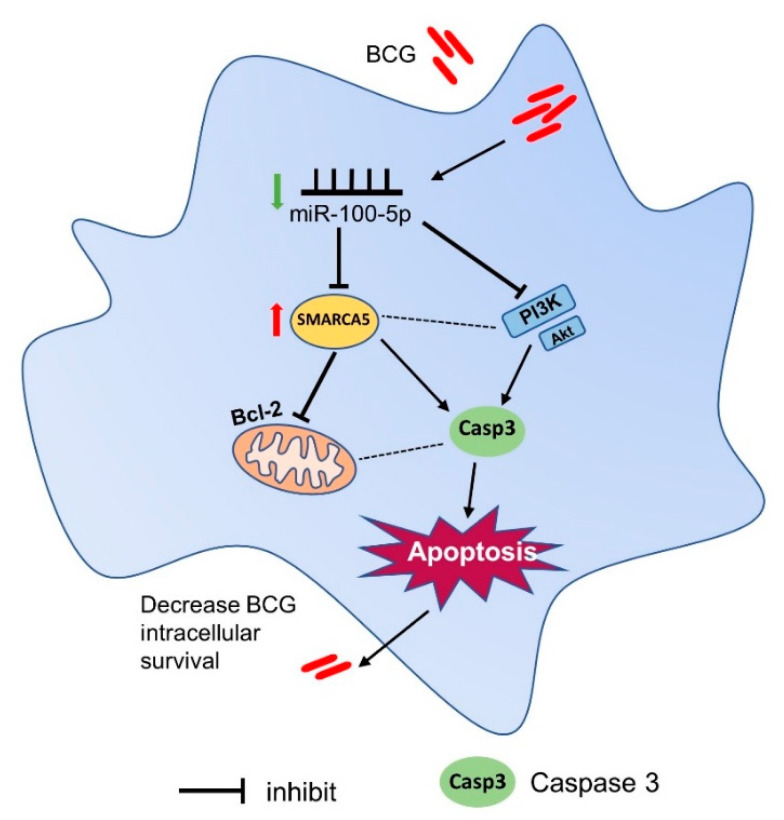
Pattern of miR-100-5p-mediated apoptosis to clear BCG in macrophages. miR-100-5p was downregulated when THP-1 cells were infected with BCG, promoting SMARCA5 expression and inhibiting the PI3K/Akt signaling pathway. Thus, caspase-3 expression increased, resulting in cell apoptosis. BCG intracellular survival was reduced finally.

**Table 1 cells-12-00476-t001:** RNA oligonucleotides sequences.

Sequence Name	Sequence 5′-3′
Has-miR-100-5p mimic negative control	UUCUCCGAACGUGUCACGUTT ACGUGACACGUUCGGAGAATT
Has-miR-100-5p mimic	AACCCGUAGAUCCGAACUUGUG CAAGUUCGGAUCUACGGGUUUU
Has-miR-100-5p inhibitor negative control	CAGUACUUUUGUGUAGUACAA
Has-miR-100-5p inhibitor	CACAAGUUCGGAUCUACGGGUU
siSMARCA5 negative control	UUCUCCGAACGUGUCACGUTT ACGUGACACGUUCGGAGAATT
siSMARCA5	AGGAAAUAUUUGAUGAUGCTT GCAUCAUCAAAUAUUUCCUCC

**Table 2 cells-12-00476-t002:** TPM value of miR-100-5p expression in RNA-seq results.

	Control	BCG-Infected	*M. tb*-Infected
6 h	4933.81	3146.33	2703.04
24 h	6258.37	2833.35	3251.83

## Data Availability

The datasets used and/or analyzed in the current study are available from the corresponding author upon reasonable request.

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
