# Peer review of "The miR-100-5p Targets SMARCA5 to Regulate the Apoptosis and Intracellular Survival of BCG in Infected THP-1 Cells"

_cells, 2023, doi:10.3390/cells12030476_

Round 1
Reviewer 1 Report
In this manuscript, the authors have explored the function of miRNA (miR-100-5p) in THP1 cells infected with BCG. The authors discovered that miR-100-5p inhibited apoptosis via regulation of SMARCA5 and PI3K/Akt signaling pathway in BCG infected THP-1 cells. However, there are some remaining questions to be answered:
1, Have the authors evaluated the function of miR-100-5p in THP1 cells without BCG infection?
2, Is the expression of miR-100-5p regulated by other infections? Or it is only specific to BCG infection?
3, The authors measured the cytotoxicity by the LDH release assay and flow cytometry, have the authors considered to measure the cell proliferation rate? Does miR-100-5p affected THP-1 cell growth induced by BCG?
4, The authors picked 5 of 7 genes in 3.5. Could the authors briefly discussed how they selected these genes?
5, what does hpi mean? Line 180, 185, etc...
6, The resolution of some figures are low and the texts are not clear to read.
Author Response
Dear reviewer,
Thank you very much for taking the time to process the submission of our original paper. We have already revised our draft. Here we simply described our revision.
Point 1: Have the authors evaluated the function of miR-100-5p in THP1 cells without BCG infection?
Response 1: Thank you for your kind comment. In the current study we didn’t evaluate the function of miR-100-5p in THP-1 cells without BCG infection. The reason is we discovered this miRNA after we infected macrophage with M.tb and BCG, those two strains are members of mycobacterium, so we want to find out the relation between miR-100-5p and mycobacteria infection. That’s why we focus on miR-100-5p after BCG infection. But we agreed that miR-100-5p in THP1 cells maybe important, too, and we will do further investigation in our following studys.
Point 2: Is the expression of miR-100-5p regulated by other infections? Or it is only specific to BCG infection?
Response 2: Thank you for your comment. In the current study we found that the expression of miR-100-5p was regulated by both BCG and M.tb infection. And according to other studies, miR-100-5p can also be regulated in other infections/disease like H5N2 virus, hepatitis B virus, osteoporosis, cardiac hypertrophy, skin cutaneous melanoma et al,. The related papers were as follows:
- Choi EJ, Kim HB, Baek YH, Kim EH, Pascua PN, Park SJ, Kwon HI, Lim GJ, Kim S, Kim YI, Choi YK. Differential microRNA expression following infection with a mouse-adapted, highly virulent avian H5N2 virus. BMC Microbiol. 2014 Sep 30;14:252. doi: 10.1186/s12866-014-0252-0. PMID: 25266911; PMCID: PMC4189662.
- Ninomiya M, Kondo Y, Kimura O, Funayama R, Nagashima T, Kogure T, Morosawa T, Tanaka Y, Nakayama K, Shimosegawa T. The expression of miR-125b-5p is increased in the serum of patients with chronic hepatitis B infection and inhibits the detection of hepatitis B virus surface antigen. J Viral Hepat. 2016 May;23(5):330-9. doi: 10.1111/jvh.12522. Epub 2016 Feb 29. PMID: 26924666.
- Wang R, Zhang M, Hu Y, He J, Lin Q, Peng N. MiR-100-5p inhibits osteogenic differentiation of human bone mesenchymal stromal cells by targeting TMEM135. Hum Cell. 2022 Nov;35(6):1671-1683. doi: 10.1007/s13577-022-00764-8. Epub 2022 Aug 10. PMID: 35947339.
- Zeng J, Wang L, Zhao J, Zheng Z, Peng J, Zhang W, Wen T, Nie J, Ding L, Yi D. MiR-100-5p regulates cardiac hypertrophy through activation of autophagy by targeting mTOR. Hum Cell. 2021 Sep;34(5):1388-1397. doi: 10.1007/s13577-021-00566-4. Epub 2021 Jun 17. PMID: 34138410.
- Zhang X, Deng Y, Liang X, Rao Y, Zheng H, Liu F, Luo X, Yang J, Chen J, Sun D. miR-100-5p Is a Novel Biomarker That Suppresses the Proliferation, Migration, and Invasion in Skin Cutaneous Melanoma. Stem Cells Int. 2022 Sep 20;2022:3585540. doi: 10.1155/2022/3585540. PMID: 36193251; PMCID: PMC9526548.
Point 3: The authors measured the cytotoxicity by the LDH release assay and flow cytometry, have the authors considered to measure the cell proliferation rate? Does miR-100-5p affected THP-1 cell growth induced by BCG?
Response 3: Thank you for your comment. In fact we measured the cell proliferation rate using CCK8 assay, and there is no significant difference among all the six groups (Figures as bellow), so we didn’t present this data in the current study.
Point 4: The authors picked 5 of 7 genes in 3.5. Could the authors briefly discussed how they selected these genes?
Response 4: Thank you very much for your comment. As we stated in the “Target gene prediction and analysis” section (Line 156-163), we used five target gene prediction sites to predict the target gene for miR-100-5p, and got seven target genes predicted by all five softwares. Among which, 6 AGO2 has been reported by other studies,so we delete it. For the other six, only five genes Luciferase reporter system were succesfully constructed. That’s how we chose the current five genes for the further study.
Point 5: what does hpi mean? Line 180, 185, etc...
Response 5: Thank you very much for thecomment. “hpi” is the abbreviation of “hours post infection”. And we explained it when it first appeared in the current study, please see the attachment.
Point 6: The resolution of some figures are low and the texts are not clear to read
Response 5: Thank you very much for your suggestion. Sorry for that, and we checked all the figures and uploaded high resolution figures again.
We really appreciated for these kind comments, thank you very much once again.
Best regards,
Li Su

Reviewer 2 Report
The work by Su L. et al is well conceived and structured. The in vitro setting is refined and different controls have been put in place. I have few minor concerns / suggestions:
1- please state the scope of this paper at the end of the introduction
2-please specify the number of replicates in the figure legends, it does ease the reading of the results in the figures
3- the discussion needs a bit of rephrasing. For a better understanding of the achievement of the paper, it would be of help to have an initial statement explaining the main results and the novelty of them before proceeding with the discussion of the results in the context of literature.
Author Response
Dear reviewer,
Thank you very much for taking the time to process the submission of our original paper. We have already revised and here we simply described our revision.
Point 1: please state the scope of this paper at the end of the introduction
Response 1: Thank you very much for this kind suggestion. We’ve added “Our findings expand the current understanding of the interaction mechanism between M. tb and macrophage. And miR-100-5p could be considered as potential targets for TB treatment “ at the end of the introduction (Line 63-65).
Point 2: please specify the number of replicates in the figure legends, it does ease the reading of the results in the figures
Response 2: Thank you very much for this careful suggestion. We specified the number of replicates in all relevant figure legends according to your suggestion.
Point 3: the discussion needs a bit of rephrasing. For a better understanding of the achievement of the paper, it would be of help to have an initial statement explaining the main results and the novelty of them before proceeding with the discussion of the results in the context of literature.
Response 3: Thank you very much for this constructive suggession. We’ve revised the description of the discussion part in revision manuscript (Line347-352).
We really appreciated for these kind comments, thank you very much once again.
Best regards,
Li Su

Round 2
Reviewer 1 Report
The authors have addressed all my questions and I have no more further concern.
Reviewer 2 Report
I thank the authors for the revision, which I believe strongly improved the quality of the paper. I recommend publication after checking of minor language imprecisions.